# Evidence against a relation between bilingualism and creativity

**Kendra V. Lange¤, Elise W. M. Hopman , Jeffrey C. Zemla, Joseph L. Austerweil** *

Department of Psychology, University of Wisconsin - Madison, Madison, WI, United States of America

¤ Current address: Department of Psychology, Pennsylvania State University, State College, PA, United States of America

* austerweil@wisc.edu

**Data Availability Statement:** All data, scripts, and related files are available from the OSF database (https://osf.io/png6a/ DOI 10.17605/OSF.IO/ PNG6A).

## Abstract

Are bilinguals more creative than monolinguals? Some prior research suggests bilinguals are more creative because the knowledge representations for their second language are similarly structured to those of highly creative people. However, there is contrasting research showing that the knowledge representations of bilinguals' second language are actually structured like those of less creative people. Finally, there is growing skepticism about there being differences between bilinguals and monolinguals on non-language tasks (e.g., the bilingual advantage for executive control). We tested whether bilinguals tested in their second language are more or less creative than both monolinguals and bilinguals tested in their first language. Participants also took a repeated semantic fluency test that we used to estimate individual semantic networks for each participant. We analyzed our results with Bayesian statistics and found support for the null hypothesis that bilingualism offers no advantage for creativity. Further, using best practices for estimating semantic networks, we found support for the hypothesis that there is no association between an individual's semantic network and their creativity. This is in contrast with published research, and suggests that some of those findings may have been the result of idiosyncrasies, outdated methods for estimating semantic networks, or statistical noise. Our results call into question reported relations between bilingualism and creativity, as well as semantic network structure as an explanatory mechanism for individual differences in creativity.

## Introduction

Scientists continue to debate whether being fluent in two or more languages hurts, helps, or has no relation to various characteristics of a person, ranging from creativity to greater financial earnings [1]. Historically, psychologists feared that bilingualism would tax the resources of the mind too much, impairing language learning and executive functioning [2, 3]. Pointing out methodological and other issues with some of the studies, other psychologists revisited the relation between bilingualism and executive functioning, finding empirical support for a "bilingual advantage" in several cognitive domains [4–6]. For example, bilinguals have better control switching between tasks [6], and a later onset of dementia than monolingual peers [5].

**Funding:** KVL was supported by the UW-Madison L&S Senior Honors Thesis Research Grant. JLA was supported by the Office of the CVGRE at UW-Madison with funding provided from the WARF. During the revision process, JLA received a research grant with American Family Insurance and started consulting for the Eureka Program. The funders had no role in the study design, data collection and analysis, decision to publish, or preparations and revision of the manuscript.

**Competing interests:** JLA received a research grant with American Family Insurance. This does not alter our adherence to PLOS ONE policies on sharing data and materials.

Recently, psychologists have questioned whether bilingualism has any effect on higher-level cognition, positive or negative [7–9]. In this paper we present a novel behavioral experiment analyzed using Bayesian hypothesis testing and find support for no relation between bilingualism and creativity.

Recently, other researchers have found empirical support that bilingual individuals are more creative than monolinguals [10–13]. Some of this research is motivated by analyzing characteristics of individuals who produce real-world creative feats. For example, nine of the top ten countries with the most Nobel prize winners per capita (in countries with populations greater than one million) have two or more official languages (Switzerland, Ireland, East Timor, and Israel) or an overwhelming majority of citizens reporting bi- or multilingualism (Sweden, Austria, Denmark, Norway, and Germany) [12]. These observational analyses are illuminating, but unfortunately they are difficult to assess from a scientific perspective due to confounds (e.g., is it multilingualism or multiculturalism?), and other concerns.

In the experimental literature, there is some support for bilinguals being more creative than monolinguals, but it is not unequivocal (e.g. [14, 15]). Further, the mechanism for this possible creativity advantage is currently not fully understood. One hypothesized mechanism is that the creativity advantage is a result of the greater cognitive control necessary for bilinguals to switch between languages [13, 16, 17]. Another hypothesized mechanism is that individual differences in creativity can be explained by differences in the manner that knowledge is stored within their semantic memory [18–20]. Thus, there is no consensus as to the mechanism responsible for individual differences in creativity. The unresolved mechanism for the bilingual advantage in creativity, combined with skepticism about the existence of bilingual advantages in general (e.g., [7–9]), prompted us to test both whether bilinguals are more creative than monolinguals or not and if so, whether the mechanism responsible for the advantage is related to how facts and knowledge are stored.

Creativity is a large concept that is often ill-defined [21]. Following researchers purporting a bilingual advantage for creativity ([10]; see [12] for a review), we focus on one aspect of creativity for the purposes of this article: The ability to come up with multiple novel solutions for a given problem. This encompasses divergent thinking—the ability to generate multiple solutions—and innovative capacity—the ability to create novel solutions. Following other researchers in this field (e.g. [13]), we use the Guilford Alternative Use Task [22] as our measure of creativity, which has participants list as many different novel uses of a given everyday object as they can think of (e.g., using a hammer as a paperweight). Thus, other aspects of creativity are beyond the scope of this article—most notably convergent thinking, the ability to solve multiple constraints, which is often tested using the Remote Associate Task (find the word common between three related words; [23]).

Using recent advances in applying network theory to understand cognition [24–26], our work focuses on the associative theory of creativity. In the associative theory [23], knowledge is structured differently in more creative individuals—it tends to be "flatter", which enables creative individuals to connect ideas across different domains. Many researchers formalize the associative theory of creativity within the framework of network theory [18, 27] by assuming that concepts in knowledge are structured according to an associative or semantic network [28]. In a semantic network, a concept is encoded as a node in the network. Concepts that are associated with one another are linked together with an edge (e.g., DOG-CAT or FIRE-RED). The nodes and edges define a network and knowledge is encoded in a distributed manner, via the patterns of associations between different concepts. See Fig 1 for a simple example of a semantic network.

Semantic networks have recently been used to explain individual differences in people's knowledge [29, 30]. In these accounts, each person has a different semantic network, which is

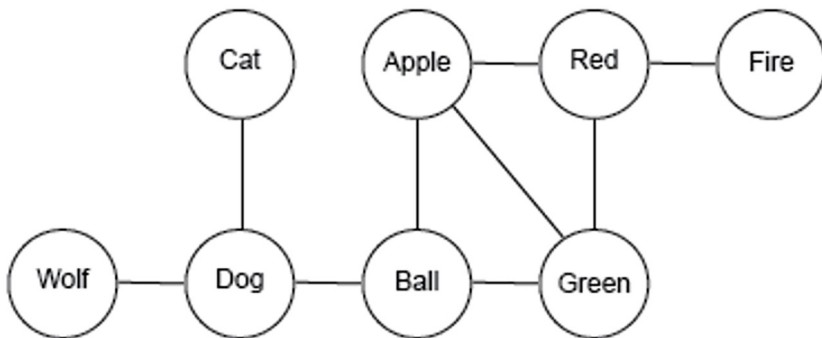

**Fig 1. Semantic networks.** An example semantic network. Concepts are encoded as nodes (circles) and associations between concepts are encoded as edges (lines). Semantic networks encode knowledge through the distributed associations between concepts.

influenced by their individual experiences with different concepts. The same cognitive processes acting on different networks can produce different behavior [31]. Thus, individual differences, such as bilingualism and creativity, may be reflected in differences in semantic networks. Previous work has found that high-creative and low-creative people have differently structured semantic networks [18, 32]. For example, a group-based semantic network estimated from high-creative individuals was more interconnected than one estimated from low-creative individuals, as measured by having a shorter average shortest path length (ASPL; the shortest path between a pair of nodes, averaged over all pairs of nodes in a network) and a larger clustering coefficient ([18]; clustering is high when two neighbors of a given node are likely to also be neighbors connected by an edge themselves). Other work has found that networks of low-creative individuals also have higher modularity, meaning the networks can be partitioned into communities (sets of nodes) that are highly interconnected [18]. Simulated search (via random walks; [33]) on the network estimated from high-creative individuals visits more unique nodes when time is limited for search [20].

In a parallel literature, researchers have also explored differences between bilinguals and monolinguals using networks. For example, Frenck-Mestre and Prince (1997) [34] proposed that bilinguals have two lexical networks, one for their first language (L1) and one for their second language (L2). Interestingly, recent work has found similarities between properties of L2 networks and networks of highly-creative individuals. Like networks of highly creative individuals, L2 networks are more connected and less modular than L1 networks [18, 35]. This implies an interesting prediction, which we coin the **representation-based advantage hypothesis**: Bilinguals should be **more creative** in their **second language** than their first due to differences in their network representations.

Although some researchers have found that the L2 semantic networks of individuals are more connected and less modular [35], other researchers found contradicting evidence. Bilson, Yoshida, Tran, Wood and Hills (2015) found that the L2 network of bilinguals had a reduced small-world index (a measure of efficient connectivity that controls for a network's likelihood to resemble a random graph of the same size [36]), and a longer ASPL than monolinguals [37]. There are a number of plausible explanations for these results: the authors tested children (6mo to 7ys), bilinguals were also found to have a sparser lexicon in both languages [37], and the learning environment for second language acquisition is often less than ideal [35]. Less than ideal language acquisition may cause suboptimal organization of the L2 network. This suboptimal organization may make information retrieval less efficient, which

could decrease creativity scores in bilinguals when tasks are conducted in their L2. This representation-based account makes the opposite prediction of the previous hypothesis and is the basis for our **representation-based disadvantage hypothesis**: Bilinguals should be **less creative** in their **second language** than their first due to differences in their network representations.

Finally, it is worth noting that there is growing skepticism in the bilingual literature about the existence of bilingual advantages [7–9]. For example, Paap, Johnson, and Sawi (2015) concluded that the proposed bilingual advantages for executive function do not exist, and that studies that demonstrate these advantages fail to match participants on demographics, have small sample sizes, or have low reproducibility [7]. Thus, we also include a **null hypothesis**: the possibility that there is no bilingual advantage or disadvantage for creativity. Further, there are methodological concerns with how researchers estimated the networks in recent network analyses of creativity. When researchers are interested in individual differences (e.g., creativity, bilingualism), the assumptions of network estimation techniques used by researchers are often violated, which can produce uninterpretable or erroneous results [38, 39]. Prior work finding a relation between network structure and creativity conducted group-based analyses using the Planar Maximally Filtered Graph network estimation method [40, 41]. The only justification for this method is its mathematical properties: It finds the largest undirected, unweighted network consistent with the correlations between participant responses that can be drawn on a plane without any of its edges intersecting. However, there is no psychological justification or validation of this method.

With respect to individual-based network analyses, to the best of our knowledge, Benedek et al. (2017) have conducted the only analysis of creativity by estimating semantic networks for each individual [27]. Although they initially attempted to use the Planar Maximally Filtered Graph method, only half of the participants had sufficient data for estimating their individual semantic network. They tried three different alternative threshold-based methods and only one of the three techniques replicated prior results showing that higher creativity is linked with more small-worldness in semantic networks.

Given these problems with previous creativity research using network analysis, we conducted a novel study that used empirically and theoretically supported best practices for estimating semantic networks. In light of our research goals including the possibility of our results supporting the null hypothesis, we use Bayesian statistics for data analyses, as these analyses are able to assess support for the null hypothesis [42].

To summarize, the predictions we identified based on prior literature for individual differences in creativity as related to bilingualism are shown in Figs 2, 3 and 4.

Note that whereas the two representation-based hypotheses only make predictions about relative differences between creativity in bilinguals tested in their L1 versus L2, the null hypothesis only makes predictions about bilinguals (irrespective of language in which they are tested) versus monolinguals. To investigate these hypotheses, we thus tested three groups of participants: English monolinguals, English-Spanish bilinguals (henceforth ES bilinguals for whom English is their first language) and Spanish-English bilinguals (henceforth SE bilinguals for whom English is their second language). To measure individual semantic networks, participants completed the repeated semantic fluency task, and we followed empirically validated best practices for estimating semantic networks from this fluency data [43]. Finally, we measured intelligence and several demographic variables, because prior work on creativity suggests that it is often correlated with intelligence and because earlier work on bilingualism has sometimes been criticized for failing to take into account demographic variables such as age, intelligence, and education level which can covary with bilingual status.

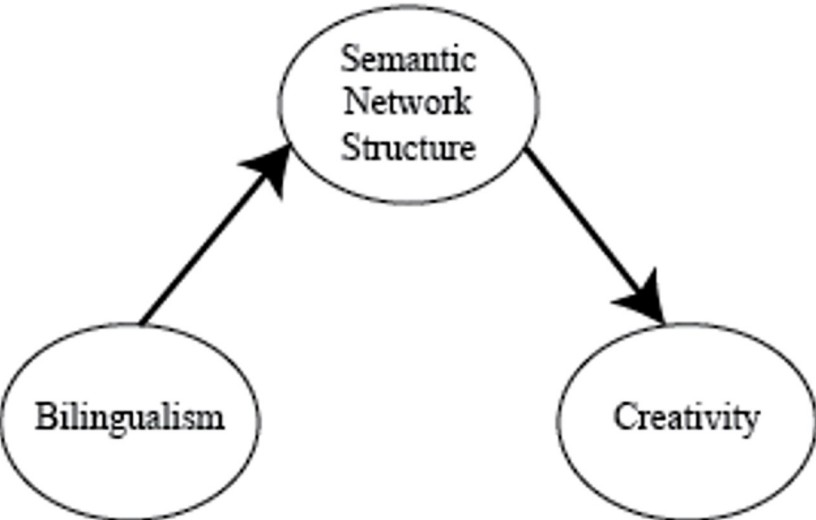

**Fig 2. Representation-based hypotheses.** Bilinguals doing creativity tasks in their second language will score higher (or lower) than bilinguals doing creativity tasks in their first language because the semantic network for a language acquired second resembles semantic networks of highly (lowly) creative individuals.

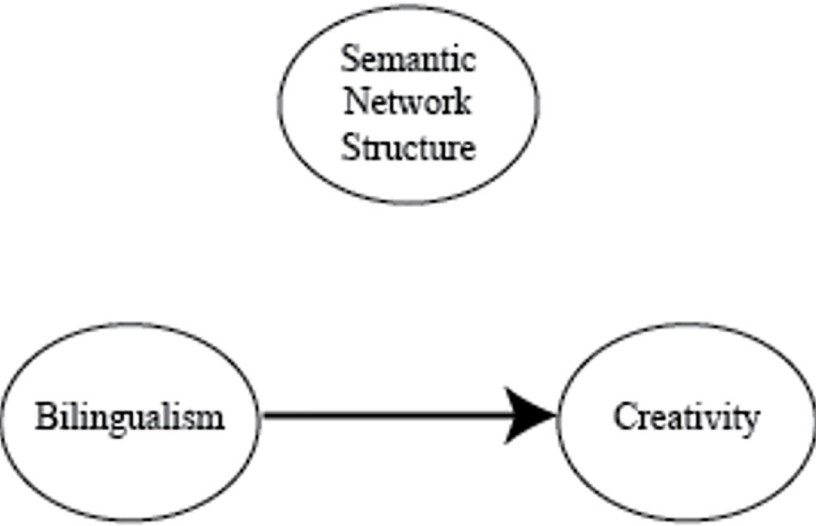

**Fig 3. Other bilingualism hypotheses.** It may be the case that factors other than differing semantic representations, such as increased executive functioning capabilities, are responsible for the relation between bilingualism and creativity. Note that we denote the causal direction from bilingualism to creativity as it is unlikely that most people become bilingual because they are creative.

## Materials and methods

### Participants

Ninety-four participants were recruited for this study from the University of Wisconsin-Madison student population. Eighty-six of these participants were recruited from the under-graduate Psychology participant pool, and 8 more participants were recruited with emails to relevant groups (e.g. student clubs related to the Spanish language) and posters around

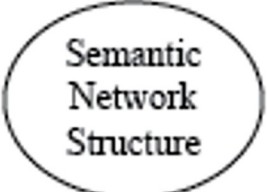

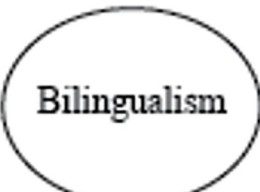
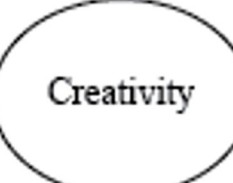

**Fig 4. Null hypothesis.** Bilingualism is unrelated to creativity.

campus. Participants were compensated for this hour-long experiment with either course credit or $10 per hour. Two of these 94 participants (one from each participant source) were excluded from all data analysis due to self-reporting proficiency in a third language other than Spanish or English.

**0.0.1 Demographics.** The final sample consisted of 92 participants (46 men, 45 women, 1 person who chose not to respond), with a mean age of 18.72 (SD = 1.08). 51 participants self-identified as white, 28 as Latinx, 7 as both white and Latinx, and the remaining 6 participants self-identified as various other (combinations of) race(s).

**0.0.2 Language screening and recruitment.** All subjects were screened before they were eligible to participate. Due to the constraints of a pre-existing language background question-naire used on the department-wide participant pool, participants were asked slightly different questions depending on whether they were recruited from this pool or not.

In the department-wide participant pool, students were eligible to participate in this study as 'monolinguals' if they indicated English as their native language and indicated no other languages learned in the home before age five. 'Bilinguals' recruited through the participant pool indicated a native language of either English or Spanish, and indicated that they had learned the other language at home before the age of five. Furthermore, these participants indicated no third language learned before the age of five.

Participants recruited outside of the department participant pool were pre-screened online using the "Language Proficiency" section of the of the Bilingual Language Profile [44]. This prescreen was used because it is detailed enough to allow us to find fluent bilingual speakers. The prescreen asked potential participants about their fluency in English and any further languages they indicated knowing in four domains: speaking, understanding, reading and writing on a scale from zero to six. Participants needed to report an average of five or above on the four questions pertaining to English to qualify for the study. Monolingual participants recruited this way also needed to report on average 1 or below on the four questions pertaining to any other language they listed to qualify for the study. Potential bilingual participants recruited this way needed to also list Spanish and report an average of 5 or above on the four questions pertaining to Spanish to qualify for the study. They also needed to report on average 1 or below for any third and further languages they listed. Their lingual status was further

tested using the Language Experience and Assessment Proficiency Questionnaire [54, 55] (described in more detail in the next section).

The vast majority of participants (85 of 92 participants) were recruited through the participant pool. Participants recruited through both methods performed similarly on our measures. However, this is difficult to assess rigorously given the small sample of participants who were recruited outside the participant pool.

## 0.1 Materials

### 0.1.1 The Guilford's Alternative Uses Task.
In order to measure creative thinking, all participants completed the Guilford's Alternative Uses task [22]. In this task, participants are instructed to type in as many alternative uses as they can think of for common object prompts like 'pencil'. Prompts were presented in two sets of three objects. Each set lasted four minutes, with participants seeing all three prompts at the same time. There was a break in between the two sets.

### 0.1.2 Raven's Progressive Matrices Test—Shortened version.
In order to control for the potential influence of intelligence on creativity scores (e.g. [16, 45, 46], participants completed the shortened version of the Raven's Progressive Matrices Test [47]. This shortened version, consisting of 12 of the original 60 questions [48], is equally reliable and much faster to complete than the complete test [49]. This is a non-verbal test of intelligence, allowing participants for whom English is a second language to perform equally well as those for whom English is their first language [50]. Participants were given no time limit for this part of the experiment.

### 0.1.3 Repeated fluency task.
To estimate individual participants' semantic networks, all participants completed a repeated semantic fluency task. In this task, they were asked to list as many animals as they could within a three minute time period [43, 51]. Participants were given this task three times during the course of the study, with at least one other task between each iteration of the fluency task. Following current best practices for estimating semantic networks of individual participants, we used U-INVITE within the SNAFU toolkit to estimate a semantic network for each individual from their fluency lists [30, 43, 52].

U-INVITE is a Bayesian inference method that finds a network that maximizes the product of two terms: the likelihood of the fluency data given a network, and the prior probability of that network. U-INVITE defines its likelihood according to a psychologically plausible retrieval model: a censored random walk [33]. In this process, an initial animal is chosen proportional to the number of semantic neighbors it has. Each subsequent animal is chosen through a random walk (i.e., choosing a semantic neighbor of the current animal at random). While an animal may appear multiple times in a random walk, only the first instance becomes part of the predicted fluency list while the remaining instances are assumed to be "censored" by an internal monitoring process. We use a large, pre-existing, group-based semantic network to estimate the prior probability of each edge in an individual's semantic network. This network was constructed from the University of South Florida (USF) free association data set [53]. In this free association task, participants were provided with a set of cue words and were asked to respond to each cue with the first word that came to mind that was meaningfully related. We constructed the network, which we call the USF network, from the data by extracting all cue-response pairs that were both animals, and adjoined them with an undirected, unweighted edge. The prior probability of an edge in a network was 2/3 if the edge exists in the USF network, and 2/5 if it does not exist in the USF network (or 0.5 if one or both of the animals were not nodes in the USF network). These parameters were previously used by Zemla and Austerweil (2019), which provides more detail on how they were determined [30]. U-INVITE conducts a stochastic search to find the network that maximizes the posterior

probability of the data. See Zemla and Austerweil (2018, 2019) and Zemla et al. (2020) for more details on this technique and the SNAFU toolkit [30, 43, 52]. For each participant network, we calculated various network characteristics like ASPL, optimal modularity, connectivity and small-world index. In cases where the estimated networks were disconnected, the largest component network was used to measure small-world index and ASPL.

**0.1.4 The Language Experience and Proficiency Questionnaire (LEAP-Q).** In order to assess participants' specific language experience, all participants completed the LEAP-Q questionnaire [54, 55]. All participants filled out the LEAP-Q about their experiences with both English and Spanish. Participants were asked, amongst other things, to rate their proficiency in speaking, understanding spoken language, and reading on a Likert scale of 1 through 10 for each language. These scores were then averaged together for each language, and participants were classified as bilingual if their self-reported Spanish and English fluency scores were both above or equal to a 6.5, in line with other bilingualism research [54, 55]. Participants also reported their age of acquisition for each language. Bilinguals who reported learning Spanish before English were classified as Spanish-English (SE) bilinguals (n = 28; 27 through the participant pool), and bilinguals who reported learning English first were classified as English-Spanish (ES) bilinguals (n = 13; nine through the participant pool). Participants who scored a two or below on their Spanish proficiency were classified as monolingual (n = 25; 23 through the participant pool). The remaining 26 participants we tested were categorized as neither bilingual nor monolingual because they scored between 2 and 6.5 on Spanish and thus fell outside of the standard criteria for mono- or bilingual on the LEAP-Q, so these participants were not included in the analyses of our main hypotheses. All participants scored above 6.5 on English, so that criterion did not lead to any exclusions.

## 0.2 Procedure

Each participant was given brief verbal instructions by an experimenter and signed a consent form before starting the experiment. Participants went through a self-guided multi-part survey through Qualtrics that included all of the different tasks mentioned above on a computer in a booth separated by a door from the experimenter. Each section of the survey was separated by a page that invited participants to take a break if they needed it. The first section consisted of demographic questions asking about age, gender identity, sexuality, and race. All participants then went through the experimental tasks in the following order: the Guilford's Alternative Uses Task, the first semantic fluency task, Raven's progressive matrices—shortened version, the second fluency task, the LEAP-Q, the final fluency task. The survey ended with a single, optional question about parental household income. We included this question because Kharkhurin (2009) found significant differences between the household incomes of bilingual and monolingual groups [10]. The full experiment was approved by the University of Wisconsin-Madison Education and Social Behavioral Sciences Institutional Review Board.

## 0.3 Coding and analysis

**0.3.1 Guilford's Alternative Uses Task coding.** Three native English speaking undergraduate research assistants scored participants' responses on the Guilford's Creativity Task. All raters were naïve to the lingual status of the participants. Raters awarded one point for every valid alternative use, with an alternative use defined as a feasible and creative way to use the object (e.g. "poking holes" as a use for a pencil). No points were granted for typical uses of the object (e.g. "writing" as a use for a pencil), or for duplicate/similar entries (e.g. a participant answering both "poke holes" and "create holes" for the prompt "pencil"). Each rater independently scored all responses. Interrater reliability between each pair of raters had a Cohen's

Kappa of 0.84 or above. Creativity scores on each item were calculated by majority consensus on a given answer, therefore if two or more raters thought an answer was creative then a point was awarded.

**0.3.2 Repeated fluency task.** As in the rest of the experiment, the repeated fluency task was conducted in English, therefore any Spanish words were counted as intrusions. Non-animal responses (e.g. 'house', 'unicorn') were also counted as intrusions. We recorded a total of 56 intrusions (<0.6%) out of 10,881 total responses across all participants. In total 22 participants (24% of participants) submitted one or more intrusions. As participants were instructed to list only animals, we removed all intrusions from the data before estimating participants' semantic networks. Likewise, all perseverations (repetitions of an animal within a list) were removed from the data before network estimation and other analyses.

**0.3.3 Statistical analysis.** We used both the frequentist null-hypothesis significance testing standard in our field and Bayesian data analyses. Unlike frequentist statistics, Bayesian data analyses can show support of the null hypothesis by quantifying the support provided for one hypothesis compared to the other by the data [56]. All Bayesian analyses were performed in JASP [57, 58], and all frequentist analyses were conducted in R [59]. For the entirety of this article, unless otherwise noted we report the Bayes Factor for the null hypothesis ($BF_{01}$) so that larger numbers denote more support for the null hypothesis. These are interpreted in the following intuitive way: if $BF_{01} = 3$ for a comparison between two groups, the data provide evidence that the null hypothesis is 3x more likely than the alternative (undirected) hypothesis that does predict a difference between the two groups. There is no standard of what BF needs to be achieved for results to be considered convincing (e.g., no equivalent of the standard frequentist criterion $p < 0.05$). Rather, researchers can decide for themselves how much more likely they want one hypothesis to be than another before they are convinced. Following Jeffreys (1961), we use the standard labels 'ambiguous' (BF around 1), 'anecdotal' ($1 < BF < 3$), 'substantial' ($3 < BF < 10$) and 'strong' ($BF > 10$) to describe our results [60]. We calculated Bayes factors to within 0.010% precision unless otherwise noted. We initially used the uniform prior that is default in JASP for all Bayesian analyses, and in cases where prior research had indicated a relationship in a certain direction, we repeated the analysis with a prior consistent with that research and report Bayes Factors for both the uniform and the directional prior.

## 0.4 Results

**0.4.1 Does bilingualism influence creativity?** *Representation-based hypotheses*. The representation-based advantage hypothesis predicts SE bilinguals will perform better than ES bilinguals on creativity tasks due to differences in their semantic networks (Fig 5, H1). Conversely, the representation-based disadvantage hypothesis predicts SE bilinguals will perform worse than ES bilinguals due to semantic network differences (Fig 5, H2). First, we assessed whether ES and SE bilinguals differed on creativity scores, using a planned frequentist mediation analysis (Fig 5). An independent samples t-test showed that SE bilinguals did not score significantly different from ES bilinguals on the creativity measure, ($M_{SE} = 16.7$, $M_{ES} = 14.5$, $N_{SE} = 28$, $N_{ES} = 13$; $t(39) = 1.0$, $p > 0.3$; Fig 6a). As there is no relationship between our two key variables of interest in our data (bilingualism and creativity), it is not meaningful to perform the subsequent steps of the mediation analysis. When we did perform the subsequent steps in the mediation, we did not find any evidence for either the positive or the negative mediation hypothesis for any network statistic tested (ASPL, modularity, connectivity, smallworldness). We then reanalyzed the creativity scores using a Bayesian independent samples t-test, finding anecdotal support that SE bilinguals and ES bilinguals had the same means on the creativity measure ($BF_{01} = 2.069$).

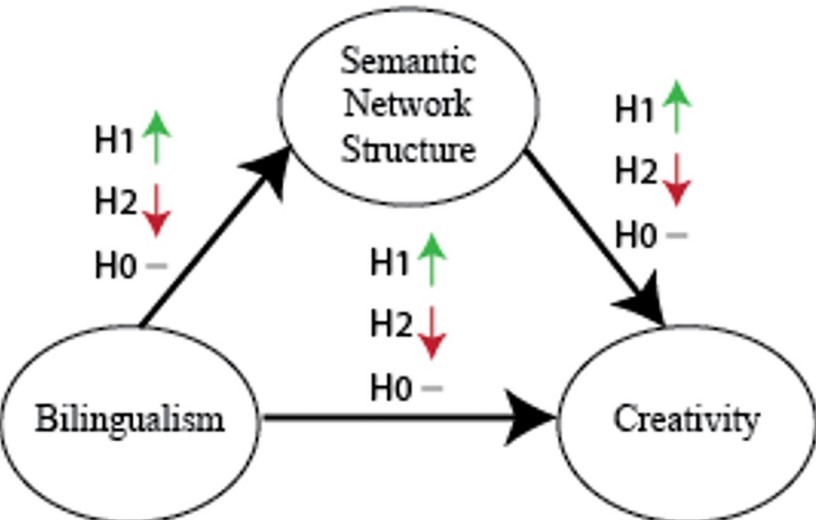

**Fig 5. Summary of the three main, competing hypotheses.** H1, the representation-based advantage hypothesis, predicts that bilinguals tested in their second language are more creative, mediated by their semantic networks having properties that are similar to the semantic networks of highly creative people. H2, the representation-based disadvantage hypothesis, predicts the opposite: namely that bilinguals tested in their second language are less creative, mediated by a network structure that is less like the networks of highly creative people. Finally, H0, the null hypothesis, predicts that there is no relationship between bilingualism and creativity.

**0.4.2 Semantic network structure in bilinguals.** Prior research has compared bilinguals on semantic fluency measures in their first and second language, so here we present our results comparing our ES and SE bilingual groups on fluency measures in the context of this prior research. In this subsection, all statistical tests are broken into SE and ES bilingual groups

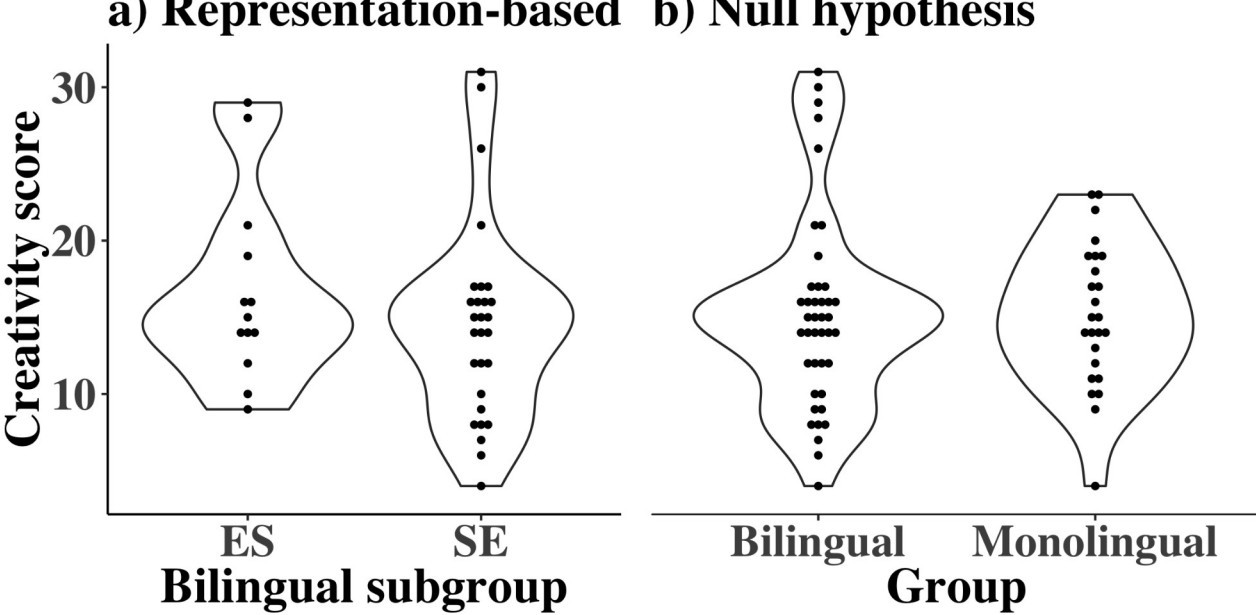

**Fig 6. Creativity and bilingualism.** Violin plots showing the distribution as well as individual scores on the creativity measure, a) to compare the two different groups of bilingual participants and b) to compare the combined bilingual group (the combination of both violin plots on the left) with the monolingual group.

($N_{SE}$ = 28 and $N_{ES}$ = 13). On the fluency measures, there was strong evidence that SE bilinguals produced fewer items than ES bilinguals ($M_{SE}$ = 33.3, $SD_{SE}$ = 6.5, $M_{ES}$ = 42.1, $SD_{ES}$ = 6.5; $BF_{10}$ = 52.5, note $BF_{10}$), which is expected given that SE bilinguals are performing the fluency task in their second language and consistent with prior findings [61]. Prior research found that bilinguals measured in their L2 (compare with our SE bilinguals) had less modular semantic networks [35]; our data are inconclusive about whether the clustering coefficients are different between ES and SE bilinguals ($BF_{01}$ = 1.06 with uniform prior; $BF_{01}$ = 0.58 with directional prior specifying lower modularity for SE bilinguals). This same study also found that bilinguals measured in their L2 had more connected semantic networks as indexed by a higher clustering coefficient; our data are inconclusive about whether modularity is different between ES and SE bilinguals ($BF_{01}$ = 1.31 with uniform prior; $BF_{01}$ = 0.73 with directional prior specifying higher clustering coefficient for SE bilinguals). Finally, that prior study found that bilinguals measured in their L2 had more connected semantic networks as indexed by a lower ASPL, in contrast with another study that found less connected semantic networks and a higher ASPL (Bilson et al., 2015); our data are inconclusive about whether ASPL is different between ES and SE bilinguals ($BF_{01}$ = 1.17 with uniform prior). The study finding less connected semantic networks also indexed this by a reduced small world coefficient; our data find substantial evidence that the small world coefficient is the same in the ES and SE bilinguals ($BF_{01}$ = 3.05 with uniform prior; $BF_{01}$ = 3.52 with directional prior specifying lower small world coefficient for SE bilinguals). Thus, overall our fluency data are inconclusive about differences between ES and SE bilinguals (for ASPL, clustering and modularity), with the exception of substantial evidence that the small world coefficient is the same in the two groups (in contrast with prior research).

**0.4.3 Individual differences in creativity.** Prior creativity research had identified that the semantic networks of more creative people are more connected as measured by a lower ASPL and a higher clustering coefficient [27]. For these and all subsequent analyses in the rest of the paper, we include the data of all 92 participants. We found substantial to strong evidence that these variables are not correlated with creativity in our sample. We found substantial evidence that the ASPL was not correlated with creativity scores ($r$ = 0.11, $BF_{01}$ = 4.65 with uniform prior, $BF_{01}$ = 14.6 with directional negative prior based on Benedek et al., 2017 and Bernard, Kenett, Ovando-Tellez, Benedek & Volle, 2019 [27, 62]; Fig 7a), and we found substantial evidence that the clustering coefficient was not correlated with creativity scores ($r$ = 0.004, $BF_{01}$ = 7.67 with uniform prior, $BF_{01}$ = 7.43 with directional positive prior based on Benedek et al., 2017 [27]; Fig 6b). Other research found that the networks of highly creative people were less modular [16], but we found substantial evidence that modularity was not correlated with creativity ($r$ = 0.12, $BF_{01}$ = 3.99 with uniform prior, $BF_{01}$ = 15.7 with directional negative prior based on Kenett et al., 2014 and Bernard et al. 2019 [18, 62]; Fig 7c). Finally, for the small world coefficient we find substantial evidence that it is uncorrelated with creativity ($r$ = 0.12, $BF_{01}$ = 3.94 with uniform prior, $BF_{01}$ = 2.25 with directional negative prior based on Bernard et al. 2019 [62]; Fig 7d).

Given the amount of support for null hypotheses we have found so far, one might be concerned about our data. However, consistent with previous work, we found a trend that intelligent people tend to be more creative ($r$ = 0.185, $p$ = 0.039 in a one-sided test expecting a positive relationship; Fig 8a; $BF_{10}$ = 1.16, note $BF_{10}$ here for an analysis to confirm the hypothesis that a positive correlation is present—encoded in the prior—based on Benedek et al., (2014) [16]). In fact, our measured correlation of 0.18 is within 0.01 of the value found in a meta-analysis [45]. In terms of network analyses we also replicate one finding we predicted based on the existing literature. Simulation research showed that networks with more unique nodes are associated with higher creativity [20]. We do find strong evidence for a correlation between creativity score and average number of items listed, which correlates with the number

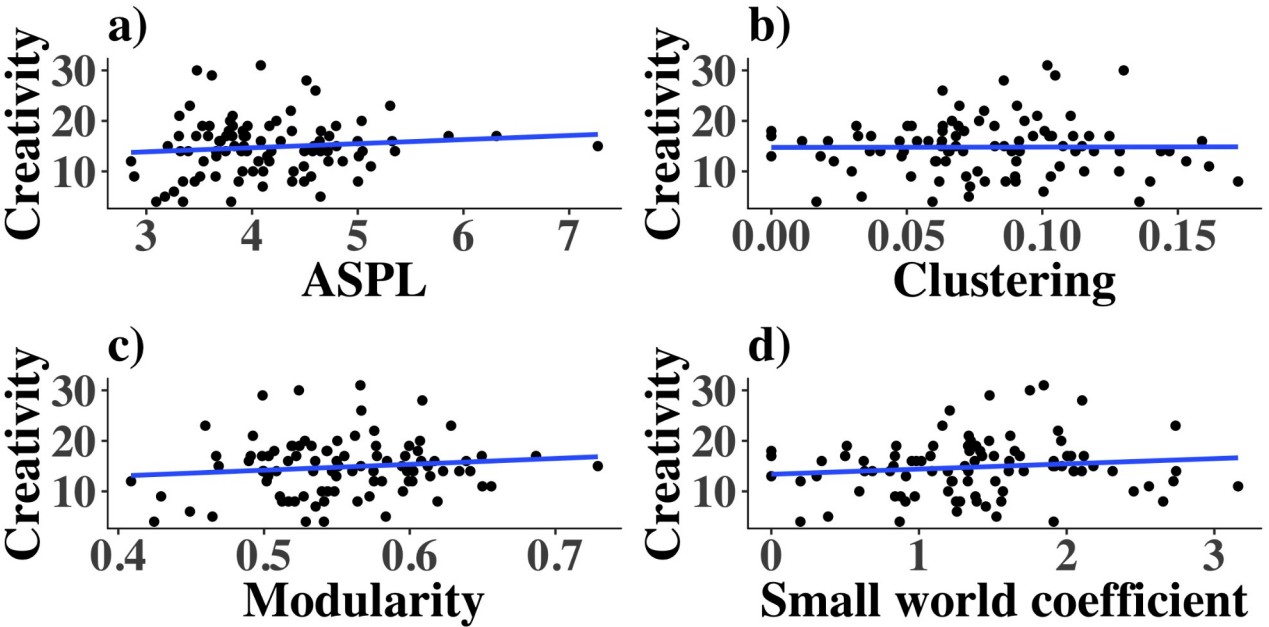

**Fig 7. Creativity and semantic network parameters.** Scatterplot with simple linear regression lines to illustrate how a) ASPL, b) clustering, c) modularity and d) small world coefficient are related to creativity for all 92 participants.

of unique nodes ($r = 0.31$, $BF_{10} = 11.5$ undirected prior; $BF_{10} = 22.9$ directed prior; Fig 8b. Note the $BF_{10}$ indicates support for the hypothesis that there is a correlation present).

**0.4.4 Ruling out alternative hypotheses.** We tested two other hypotheses to rule out alternative hypotheses.

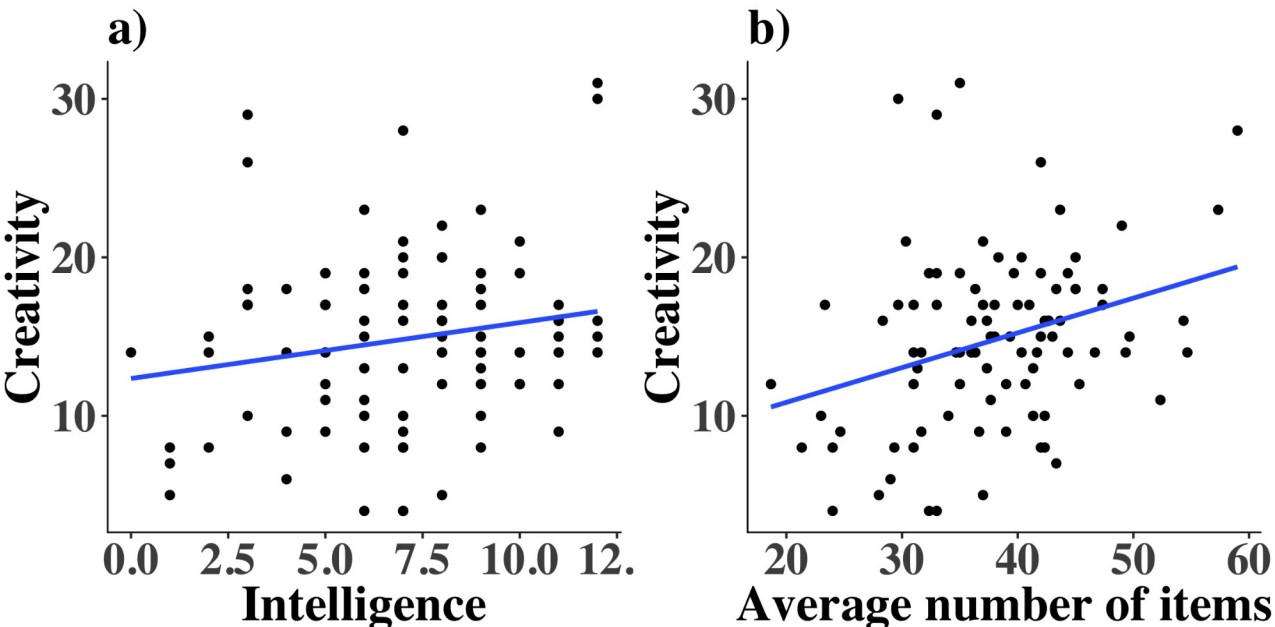

**Fig 8. Replicating prior results.** Scatterplot with a simple linear regression line to illustrate how a) intelligence and b) number of items listed in the fluency task are related to creativity for all 92 participants.

*Native versus non-native speakers.* In our experiment, SE bilinguals are performing all tasks in their second language, and ES bilinguals as well as monolinguals were performing the task in their native language. We analyzed whether this resulted in any differences. Thus ES bilinguals and monolinguals were grouped into a "native English group" (N = 38) and compared to SE bilinguals who became the "native Spanish group" (N = 28). We found substantial evidence that native English speakers performed equally well as the native Spanish speakers on the creativity measure ($BF_{01}$ = 4.17). As our main question hinges on the creativity results and these do not seem affected by native language, we concluded that performance on the creativity task was not driven by second language difficulty.

*Demographic differences between groups.* Kharkhurin (2009) found significant differences in socioeconomic status of bilingual versus monolingual respondents, and speculated that this may have affected creativity scores between the groups [10]. However, we found anecdotal evidence for similarity in self-reported family income (of those who chose to report) between ES versus SE bilinguals ($BF_{01}$ = 2.15), ES bilinguals versus monolinguals ($BF_{01}$ = 2.23), and SE bilinguals and Monolinguals ($BF_{01}$ = 2.83). A chi-squared analysis indicates that the number of participants per group choosing to respond is not different between groups, $p > 0.6$. It is the last score that is most informative, as Kharkhurin (2009)'s bilingual group was also tested in their second language, and thus most comparable to our SE bilingual group [10].

## 1 Discussion

Our research failed to replicate previous work on the bilingual advantage or disadvantage on creativity, and supports the null hypothesis that bilingualism is not associated with creativity. We found no support for a representation-based advantage hypothesis or a representation-based disadvantage hypothesis. ES and SE bilinguals neither differed substantially nor did we have substantial evidence that their creativity scores were the same. Monolinguals scored as high on our test of creativity as bilinguals taken as one group. This is surprising given previous research [10] on the bilingual creative advantage but does align with recent claims against the bilingual advantage more generally [7, 8].

We hypothesized that semantic networks could account for differences in creative thinking. Previous research indicated that the connectivity of a semantic network helped people think creatively, with more creative people having a more connected semantic network [18, 35]. However, only one measure from the semantic fluency task turned out to be meaningfully correlated with creativity in our data: the average number of items listed. This is not surprising: in these two tasks participants are prompted to list as many animals or alternative uses as possible in a short time frame. The correlation between these measures suggests that people's ability to list responses quickly is more important than characteristics of the semantic network per se. If it were driven by differences in knowledge, than they are not captured by simple statistics of semantic networks. Although the ASPL was unambiguously not correlated with creativity, the lack of other substantial correlations between creativity scores and network measures like modularity and clustering coefficient indicate that semantic networks differences do not account for differences in creative thinking. This fails to replicate prior results supporting theories that pose that the structural properties of semantic networks influence creativity [18, 63].

Our data were ambiguous as to evidence that SE and ES bilinguals were more or less creative. While we did find that they differed in the number of items listed in a fluency task, this was an expected result as there may be differences in speed of processing or the English vocabulary size between native and non-native English speakers. This again casts doubt on previous research and both of the representation-based hypotheses. Unsurprisingly, when we combined ES and monolinguals together to form a native English speakers group we found they were

able to list more words on the fluency task than native Spanish speakers. Though the ability to list words correlated with creativity, native English speakers had no benefit in the creativity measure. In fact, proficiency in English did not correlate at all with the ability to list words on a fluency task or with creativity. This may be due to ceiling effects, as all Spanish native speakers in our experiment are living in an English speaking country and therefore can be presumed to get plenty of practice in their second language. Though our ES bilingual sample size was small, our sampling was relatively consistent between groups: gender ratios differed between groups, but age and SES did not.

## 2 Limitations

We had a small sample size of ES bilinguals (N = 13), which reduced the statistical power of some aspects of the study. In particular, this limits the strength of the conclusions related to differences between L1 and L2 of bilinguals. However, almost all of our results with ES bilinguals had anecdotal or stronger support for the null hypothesis, namely there being no relation between creativity and bilingualism. Conclusions related to the general difference between bilinguals and monolinguals are stronger due to their larger sample sizes (41 and 25, respectively). The strongest conclusions are those that tested individual differences in estimated semantic networks as those included 92 participants. Further, we did replicate previous research finding a relation between intelligence and creativity. This mitigates concern that we did not find a result that appeal to our experiment having insufficient power to find any result.

Another limitation is that bilinguals are a very diverse group of people with incredibly different characteristics. Therefore our findings on a group of undergraduate participants may not represent trends found in bilinguals as a whole. However, given our long study time (40 minutes to an hour), we thought we could engage participants more if participants were run in person as opposed to recruiting a more diverse sample online through MTurk or another online survey. Furthermore, other factors besides language ability are relevant when looking at bilingual groups. Cultural background has been shown to be relevant in creativity research [64]. While studying only Spanish-English bilinguals helped us reduce some of the possible variability due to cultural background, these bilinguals are only a small subset of the bilingual population in the US, and may not be representative of bilinguals as a whole. For example, some prior work reporting a bilingual advantage had Hebrew-English and other types of bilinguals as participants [18, 27]. It is possible that this advantage generalizes across some bilinguals, but not to Spanish-English bilinguals. Finally, we focus on semantic networks for animals, which is a standard domain for examining differences in knowledge within the semantic cognition literature. It is unclear whether these results generalize across human knowledge.

## 3 Conclusion

Our study is the first to find support for there being no relation between bilingualism and creativity. Using best practices for estimating semantic networks, we found support that there is no relation between the structure of a person's semantic network and the extent of their creativity. This contradicts some recent work in this area, and suggests that researchers finding a relation between semantic network structure and creativity may have done so due to methodological issues or idiosyncrasies. The question is far from closed and more research in the area needs to be conducted. However, without publishing null results, especially those that find substantive support for the null hypothesis, researchers will be unable to appropriately judge whether there are relations [65, 66]. Thus, our work contributes to the growing body of research questioning relations between bilingualism and different cognitive and

other psychological factors, and is essential for ensuring that these important questions are tested in a rigorous manner.

## Acknowledgments

We thank Elizabeth Pettit and Anantha Rao for help scoring the Guilford's responses. We thank current and former Austerweil lab members, as well as the audience at CogSci 2018, for feedback on this study.

## Author Contributions

**Conceptualization:** Kendra V. Lange, Joseph L. Austerweil.

**Data curation:** Kendra V. Lange, Elise W. M. Hopman, Joseph L. Austerweil.

**Formal analysis:** Kendra V. Lange, Elise W. M. Hopman, Jeffrey C. Zemla, Joseph L. Austerweil.

**Funding acquisition:** Kendra V. Lange.

**Investigation:** Kendra V. Lange, Joseph L. Austerweil.

**Methodology:** Kendra V. Lange, Elise W. M. Hopman, Joseph L. Austerweil.

**Project administration:** Kendra V. Lange, Joseph L. Austerweil.

**Supervision:** Elise W. M. Hopman, Joseph L. Austerweil.

**Validation:** Kendra V. Lange, Elise W. M. Hopman, Joseph L. Austerweil.

**Visualization:** Kendra V. Lange, Elise W. M. Hopman, Joseph L. Austerweil.

**Writing – original draft:** Kendra V. Lange, Elise W. M. Hopman, Jeffrey C. Zemla, Joseph L. Austerweil.

**Writing – review & editing:** Kendra V. Lange, Elise W. M. Hopman, Jeffrey C. Zemla, Joseph L. Austerweil.

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
