## [Decision Letter · Decision Letter 0]

30 Apr 2020

PONE-D-20-03820

Evidence against the bilingual advantage for creativity and the semantic network theory of creativity

PLOS ONE

Dear Dr Austerweil,

Thank you for submitting your manuscript to PLOS ONE. First, let me apologise for the the lengthy review time. Given the current pandemic, finding reviewers was not easy.  I'm now quite pleased to be able to write this letter. After careful consideration from two reviewers and myself, we feel that it has merit but does not fully meet PLOS ONE’s publication criteria as it currently stands. Therefore, we invite you to submit a revised version of the manuscript that addresses the points raised during the review process.

I'm confident that you can address the reviewers' comments below. I have one additional comment. In the introduction, it would be good to address whether there is evidence that bilingualism increases creativity based on analysis of real-world creative achievement. Or does all the evidence come from increased performance on cognitive tasks meant to measure component cognitive processes that support real-world creativity? For example, are bilingual painters/musicians/authors more successful by some metric than monolingual painters/musicians/authors? There is evidence that international travel in scientists is associated with greater citation rate, and immigrants are over-represented as Nobel Prize winners. Are there similar analyses focusing on bilingualism specifically? Of course these kinds of observational studies have confounds that the kinds of experiments conducted here do not, but it is good to set the stage of what these experiments are trying to explain by discussing the pattern in the real-world. If there is no such evidence that bilinguals on the whole produce greater creative achievements, then it puts the search for the effects in experimental contexts in a different light. I imagine some of the papers you cited currently would directly address this issue. 

We would appreciate receiving your revised manuscript by Jun 14 2020 11:59PM. To enhance the reproducibility of your results, we recommend that if applicable you deposit your laboratory protocols in protocols.io, where a protocol can be assigned its own identifier (DOI) such that it can be cited independently in the future. For instructions see: http://journals.plos.org/plosone/s/submission-guidelines#loc-laboratory-protocols

We look forward to receiving your revised manuscript.

Kind regards,

Micah B. Goldwater, Ph.D

Academic Editor

PLOS ONE

Journal Requirements:

"KVL was supported by the UW-Madison L&S Senior Honors Thesis Research Grant. JLA was supported by the Office of the CVGRE at UW-Madison with funding provided from the WARF."

"The funders had no role in study design, data collection and analysis, decision to

publish, or preparation of the manuscript."

Reviewers' comments:

Reviewer's Responses to Questions

**Comments to the Author**

1. Is the manuscript technically sound, and do the data support the conclusions?

Reviewer #1: Yes

Reviewer #2: Yes

2. Has the statistical analysis been performed appropriately and rigorously? 

Reviewer #1: Yes

Reviewer #2: Yes

3. Have the authors made all data underlying the findings in their manuscript fully available?

Reviewer #1: No

Reviewer #2: Yes

4. Is the manuscript presented in an intelligible fashion and written in standard English?

Reviewer #1: No

Reviewer #2: Yes

5. Review Comments to the Author

Reviewer #1: Overall an article that is sound and rigorous that could benefit with a few notes.

Re: #3: Data is not currently available on receipt of the manuscript for review, but authors do indicate that it will be available upon acceptance.

Re: #4: Minor errors throughout; interleaved with additional notes below for clarity.

line 28-30: Important to specify that flexible thinking is only one aspect of creativity; also, along with the demographic information collected, I would have liked to see educational background, since some college majors (for example) may have more experience with the type of flexible thinking that is being tested here.

line 120-121: The line about the H0 seems out of place here; move it up to at least the start of the paragraph, if not the start of the section on your hypotheses.

line 170: Include a line about how different the participants recruited via subject pool and those recruited and given the questionnaire to determine their bilingual status were.

line 262: change "...household incomes bilingual and monolingual" to "...household incomes OF bilingual and monolingual"

line 265: Remove "Plan" from the section heading; it is no longer a plan since the analysis has been done.

line 282: fix Latex typo at the beginning of the line

line 283: spell out "1" for consistency

line 345: remove "prior" to reduce any references to non-Bayesian priors

line 382: opening parenthesis needs to be closed somewhere

line 383, 391: Unclear whether BF10 is a typo for BF01 as used previously

line 392: Tense/aspect of "indicating" is odd; consider "indicates"

line 414: add "or disadvantage", since your data also doesn't support that bilingualism (or at least being tested in the L2) confers any disadvantage on flexible thinking. To that note, your title is splashy but somewhat misleading, since you find no advantage or disadvantage for bilinguals and their creativity, narrowly defined.

line 446-448: In reading the rest of the article, it's unclear to me "how bilingual" the SE and ES bilinguals were—that is, what does the various questionnaires' scoring really indicate regarding their language use and proficiency? If they had to name animals in English for the fluency task and all were effectively native-like speakers of English, I'm not sure the fluency task is relevant or meaningful (and is indeed likely a ceiling effect as you write); this could be different if there were more variability in the proficiency levels of both sets of bilinguals.

line 468-469: Take the ref to the studies about Hebrew out of the parenthetical, since that helps support your point.

Finally, are the lexicosemantic neighborhoods similar between English and Spanish? Differences in mapping can show up in word use (e.g., Bowerman & Choi 2001, Choi 2006, Pavlenko & Malt 2011), which might influence the intrusions or general performance on the fluency task.

Reviewer #2: Overall this paper provides a valuable contribution to the study of how bilingualism may affect general cognitive abilities, and creativity specifically. The authors methods are sound, as is how they theoretically situate their research. Specifically, the use of bayesian statistics to quantify evidence for null results is appropriate for the aims of the paper.

The following are some points I believe need addressing:

1. A notable limitation of the empirical study is the sample size, specifically the sample of only 13 bilinguals for whom Spanish is their second language and English their first (the ES sample). The authors do address this limitation in section 2 (line 451). However, there is a nuance here that I believe needs further clarification. The authors provide a defence against the claim that the sample was underpowered by appealing to the fact that they replicated a relation between intelligence and creativity. My issue is that this replication was for the whole sample of 92 participants (unless I am mistaken), which seems to me orthogonal to the question of whether the ES sample is sufficient to make the various comparisons to the SE and monolingual samples. Thus, a clearer link needs to be drawn to justify the connection or this needs to be separated from the defence of the ES sample size limitation.

Furthermore, the authors also defend against this ES sample size limitation by referring to the fact that results including the ES sample showed anecdotal or stronger support for the null. Again here, there is potential for confusion as the ‘better than anecdotal’ level evidence was only for when the ES sample was combined with the larger SE sample and then compared against monolinguals. There were also a number of inconclusive comparisons (between SE and ES) regarding semantic network parameters not mentioned here in the limitations section, which should be mentioned as results which may have been limited by sample size.

In summary, I feel that this limitations sections should be revised to make more transparent which particular conclusions from the analysis were limited by this small ES sample.

Additionally, there are a number of other small modifications regarding typos and general style etc:

2. The introduction section is well written, however, the section from line 1 to line 68 needs to be broken up into smaller paragraphs, as it presently appears to be a single massive one. For example, I recommend a paragraph break after “creativity” on line 14 and again after “stored” on line 28… as well as further breaks in this vein going forward.

3. There is no author information on reference 9 (i.e. Kharkhurin, 2009 reference), also no journal information on ref 26. Please double check other references as well.

4. Please remove fullstops at the beginning of paragraphs on line 267, 279, 288, 335, 362.

5. No space after fullstop on line 419 e.g. “group.This” also no space in “hypothesis,namely” on 454

6. You may wish to include a recently published large sample demonstration of no general advantage of bilingualism - Nichols, E. S., Wild, C. J., Stojanoski, B., Battista, M. E., & Owen, A. M. (2020). Bilingualism affords no general cognitive advantages: a population study of executive function in 11,000 people. Psychological Science. https://doi.org/10.1177/0956797620903113 …. e.g. this may be inserted somewhere around line 96

7. I recommend rewriting the sentences containing bracketed stats in the following way, such that the bracketed stats are at the end of the phrase. This makes it far easier to read. e.g. your original was “We found substantial evidence that the ASPL (r = 0.11, BF01 = 4.65 with uniform prior, BF01 = 14.6 with directional negative prior based on Benedek et al., 2017 and Bernard, Kenett, Ovando-Tellez, Benedek & Volle, 2019 [26,59]; Fig 7a) was not correlated with creativity scores.” —> I recommend “We found substantial evidence that the ASPL was not correlated with creativity scores (r = 0.11, BF01 = 4.65 with uniform prior, BF01 = 14.6 with directional negative prior based on Benedek et al., 2017 and Bernard, Kenett, Ovando-Tellez, Benedek & Volle, 2019 [26,59]; Fig 7a). ”

Finally, I thank the authors for their hard work and hope they are keeping well in this current stressful covid-19 situation.

6. PLOS authors have the option to publish the peer review history of their article (what does this mean?). If published, this will include your full peer review and any attached files.

Reviewer #1: Yes: Shiloh Drake

Reviewer #2: Yes: Courtney B Hilton

---

## [Author Response · Author response to Decision Letter 0]

28 May 2020

Dear Dr Austerweil,

Thank you for submitting your manuscript to PLOS ONE. First, let me apologise for the the lengthy review time. Given the current pandemic, finding reviewers was not easy. I'm now quite pleased to be able to write this letter. After careful consideration from two reviewers and myself, we feel that it has merit but does not fully meet PLOS ONE’s publication criteria as it currently stands. Therefore, we invite you to submit a revised version of the manuscript that addresses the points raised during the review process.

I'm confident that you can address the reviewers' comments below. 

Thank you and the reviewers for your hard work reviewing our manuscript. We understand that it is a difficult time. We especially appreciate the detailed feedback clarifying our writing. It will strengthen the manuscript.

I have one additional comment. In the introduction, it would be good to address whether there is evidence that bilingualism increases creativity based on analysis of real-world creative achievement. Or does all the evidence come from increased performance on cognitive tasks meant to measure component cognitive processes that support real-world creativity? For example, are bilingual painters/musicians/authors more successful by some metric than monolingual painters/musicians/authors? There is evidence that international travel in scientists is associated with greater citation rate, and immigrants are over-represented as Nobel Prize winners. Are there similar analyses focusing on bilingualism specifically? Of course these kinds of observational studies have confounds that the kinds of experiments conducted here do not, but it is good to set the stage of what these experiments are trying to explain by discussing the pattern in the real-world. If there is no such evidence that bilinguals on the whole produce greater creative achievements, then it puts the search for the effects in experimental contexts in a different light. I imagine some of the papers you cited currently would directly address this issue. 

We agree adding an example related to real-world creativity to the introduction strengthens our manuscript. We added the following paragraph to the introduction which appears on page 2 of the revised manuscript. 

Some of this research is motivated by analyzing characteristics of individuals who produce real-world creative feats. For example, nine of the top ten countries with the most Nobel prize winners per capita (in countries with populations greater than one million) have two or more official languages (Switzerland, Ireland, East Timor, and Israel) or an overwhelming majority of citizens reporting bi- or multilingualism (Sweden, Austria, Denmark, Norway, and Germany) (Kharkhurin, 2012). These observational analyses are illuminating, but unfortunately, they are difficult to assess from a scientific perspective due to confounds (e.g., is it multilingualism or multiculturalism?), and other concerns. 

See page one of the cover letter.

To enhance the reproducibility of your results, we recommend that if applicable you deposit your laboratory protocols in protocols.io, where a protocol can be assigned its own identifier (DOI) such that it can be cited independently in the future. For instructions see: http://journals.plos.org/plosone/s/submission-guidelines#loc-laboratory-protocols

Experimental results and statistical analyses are available on OSF: https://osf.io/png6a/ (DOI 10.17605/OSF.IO/PNG6A)

They are accessible now at https://osf.io/png6a/

"KVL was supported by the UW-Madison L&S Senior Honors Thesis Research Grant. JLA was supported by the Office of the CVGRE at UW-Madison with funding provided from the WARF."

"The funders had no role in study design, data collection and analysis, decision to

publish, or preparation of the manuscript."

We cut the funding information from the acknowledgements. Please see page one of the cover letter for the updated text of the Funding Statement. 

Reviewers' comments:

Reviewer #1: Overall an article that is sound and rigorous that could benefit with a few notes.

Re: #3: Data is not currently available on receipt of the manuscript for review, but authors do indicate that it will be available upon acceptance.

They are now available at https://osf.io/png6a/.

Re: #4: Minor errors throughout; interleaved with additional notes below for clarity.

line 28-30: Important to specify that flexible thinking is only one aspect of creativity; 

Thank you for the suggestion. We added a clarification to the revised manuscript, which appears on lines 37-40.

also, along with the demographic information collected, I would have liked to see educational background, since some college majors (for example) may have more experience with the type of flexible thinking that is being tested here.

Unfortunately, we did not record the participants’ majors. We agree educational background could be illuminating for the reasons the reviewer mentions.

line 120-121: The line about the H0 seems out of place here; move it up to at least the start of the paragraph, if not the start of the section on your hypotheses.

Thank you for this suggestion. We moved the definition of the null hypothesis to earlier in our section on hypotheses. It now appears on lines 108-110 of the revised manuscript.

line 170: Include a line about how different the participants recruited via subject pool and those recruited and given the questionnaire to determine their bilingual status were.

Added to manuscript “The vast majority of participants (85 of 92 participants) were recruited through the participant pool. Participants recruited through both methods performed similarly on our measures. However this is difficult to assess rigorously given the small sample of participants who were recruited outside the participant pool.” It appears on line 

line 262 line 382: opening parenthesis needs to be closed somewhere

 Thank you. We implemented these minor changes throughout.

line 383, 391: Unclear whether BF10 is a typo for BF01 as used previously

 It is not a typo. Thank you for checking.

line 392: Tense/aspect of "indicating" is odd; consider "indicates"

 Thank you. We implemented this change which appears on line 402.

line 414: add "or disadvantage", since your data also doesn't support that bilingualism (or at least being tested in the L2) confers any disadvantage on flexible thinking. To that note, your title is splashy but somewhat misleading, since you find no advantage or disadvantage for bilinguals and their creativity, narrowly defined.

We agree and decided to change our title to: Evidence against a relation between bilingualism and creativity

line 446-448: In reading the rest of the article, it's unclear to me "how bilingual" the SE and ES bilinguals were—that is, what does the various questionnaires' scoring really indicate regarding their language use and proficiency? If they had to name animals in English for the fluency task and all were effectively native-like speakers of English, I'm not sure the fluency task is relevant or meaningful (and is indeed likely a ceiling effect as you write); this could be different if there were more variability in the proficiency levels of both sets of bilinguals.

The bilingualism literature has wrestled for a long time with respect to how to define a “true” bilingual and different gradations from there. The measure we use, the LEAP-Q, is a well-established method in the literature. There is no component of the LEAP-Q that elicits participants to list animals in any language. Assessing the efficacy of the LEAP-Q and what the best measure of bilingualism proficiency for examining differences in creativity between monolinguals and bilinguals is beyond the scope of our manuscript.

line 468-469: Take the ref to the studies about Hebrew out of the parenthetical, since that helps support your point.

This suggestion is implemented in the revised manuscript on line 407.

Finally, are the lexicosemantic neighborhoods similar between English and Spanish? Differences in mapping can show up in word use (e.g., Bowerman & Choi 2001, Choi 2006, Pavlenko & Malt 2011), which might influence the intrusions or general performance on the fluency task.

There are many potential lexiosemantic differences to investigate between English and Spanish. We agree with the reviewer that it would be interesting to examine them and their relation to fluency tasks. However, the number of errors in the fluency data is quite small (only 56 of 10,881 responses). Given the small number of errors, it would be extremely challenging to make any statistically meaningful claims about errors on the fluency task and the lexiosemantic properties of English and Spanish from our results.. 

Reviewer #2: Overall this paper provides a valuable contribution to the study of how bilingualism may affect general cognitive abilities, and creativity specifically. The authors methods are sound, as is how they theoretically situate their research. Specifically, the use of bayesian statistics to quantify evidence for null results is appropriate for the aims of the paper.

The following are some points I believe need addressing:

1. A notable limitation of the empirical study is the sample size, specifically the sample of only 13 bilinguals for whom Spanish is their second language and English their first (the ES sample). The authors do address this limitation in section 2 (line 451). However, there is a nuance here that I believe needs further clarification. The authors provide a defence against the claim that the sample was underpowered by appealing to the fact that they replicated a relation between intelligence and creativity. My issue is that this replication was for the whole sample of 92 participants (unless I am mistaken), which seems to me orthogonal to the question of whether the ES sample is sufficient to make the various comparisons to the SE and monolingual samples. Thus, a clearer link needs to be drawn to justify the connection or this needs to be separated from the defence of the ES sample size limitation.

Furthermore, the authors also defend against this ES sample size limitation by referring to the fact that results including the ES sample showed anecdotal or stronger support for the null. Again here, there is potential for confusion as the ‘better than anecdotal’ level evidence was only for when the ES sample was combined with the larger SE sample and then compared against monolinguals. There were also a number of inconclusive comparisons (between SE and ES) regarding semantic network parameters not mentioned here in the limitations section, which should be mentioned as results which may have been limited by sample size.

In summary, I feel that this limitations sections should be revised to make more transparent which particular conclusions from the analysis were limited by this small ES sample.

Thank you for pointing this out. We agree this was unclear. The Ns are now explicitly stated in each subsection of the analyses. We also revised the limitations to be more precise about the particular conclusions affected by the small ES sample. The changes to the limitations section appear on lines 465-469 of the revised manuscript.

Additionally, there are a number of other small modifications regarding typos and general style etc:

2. The introduction section is well written, however, the section from line 1 to line 68 needs to be broken up into smaller paragraphs, as it presently appears to be a single massive one. For example, I recommend a paragraph break after “creativity” on line 14 and again after “stored” on line 28… as well as further breaks in this vein going forward.

Thank you for pointing this out. This was a mistake due to typesetting as we went through the submission process. We have broken the introduction into several paragraphs.

3. There is no author information on reference 9 (i.e. Kharkhurin, 2009 reference), also no journal information on ref 26. Please double check other references as well.

4. Please remove fullstops at the beginning of paragraphs on line 267, 279, 288, 335, 362.

 Fixed. Thank you.

5. No space after fullstop on line 419 e.g. “group.This” also no space in “hypothesis,namely” on 454

 Fixed. Thank you.

6. You may wish to include a recently published large sample demonstration of no general advantage of bilingualism - Nichols, E. S., Wild, C. J., Stojanoski, B., Battista, M. E., & Owen, A. M. (2020). Bilingualism affords no general cognitive advantages: a population study of executive function in 11,000 people. Psychological Science. https://doi.org/10.1177/0956797620903113 …. e.g. this may be inserted somewhere around line 96

Thank you for the reference -- We have added it to the manuscript in several places, including line 11.

7. I recommend rewriting the sentences containing bracketed stats in the following way, such that the bracketed stats are at the end of the phrase. This makes it far easier to read. e.g. your original was “We found substantial evidence that the ASPL (r = 0.11, BF01 = 4.65 with uniform prior, BF01 = 14.6 with directional negative prior based on Benedek et al., 2017 and Bernard, Kenett, Ovando-Tellez, Benedek & Volle, 2019 [26,59]; Fig 7a) was not correlated with creativity scores.” —> I recommend “We found substantial evidence that the ASPL was not correlated with creativity scores (r = 0.11, BF01 = 4.65 with uniform prior, BF01 = 14.6 with directional negative prior based on Benedek et al., 2017 and Bernard, Kenett, Ovando-Tellez, Benedek & Volle, 2019 [26,59]; Fig 7a). ”

Thank you for the suggestion. We agree this formatting is more clear. It has been updated in the revised manuscript on lines 375-389 (and elsewhere).

Finally, I thank the authors for their hard work and hope they are keeping well in this current stressful covid-19 situation.

We thank the reviewer for their thoughtfulness and kind thoughts. We hope the same for them. 

6. PLOS authors have the option to publish the peer review history of their article (what does this mean?). If published, this will include your full peer review and any attached files.

In a desire to be as transparent as possible, we would like to publish the peer review history of our article.

---

## [Editor Report · Decision Letter 1]

5 Jun 2020

Evidence against a relation between bilingualism and creativity

PONE-D-20-03820R1

Dear Dr. Austerweil,

We’re pleased to inform you that your manuscript has been judged scientifically suitable for publication and will be formally accepted for publication once it meets all outstanding technical requirements.

Kind regards,

Micah B. Goldwater, Ph.D

Academic Editor

PLOS ONE
---

## [Editor Report · Acceptance letter]

10 Jun 2020

PONE-D-20-03820R1 

Evidence against a relation between bilingualism and creativity 

Dear Dr. Austerweil:

I'm pleased to inform you that your manuscript has been deemed suitable for publication in PLOS ONE. Congratulations! Your manuscript is now with our production department. 

Kind regards, 

on behalf of

Dr. Micah B. Goldwater 

Academic Editor

PLOS ONE